# Comparison of unbiased metagenomic next generation sequencing to targeted multiplex diagnostic assays for the detection of respiratory viruses

Justin Hardick[1☺], Raghavendran Anantharam[1☺], Jennifer Lu[2,3,4], Steven L. Salzberg[2,3,5], Richard E. Rothman[6], Katherine Z. J. Fenstermacher[6], Andrew Pekosz[7], Annet Onzia[8], Lydia Nakiyingi[8], Yukari C. Manabe[1‡], Abraham J. Kandathil[1‡]*

1 Division of Infectious Diseases, Johns Hopkins School of Medicine, Baltimore, Maryland, United States of America, 2 Center for Computational Biology, Johns Hopkins University, Baltimore, Maryland, United States of America, 3 Department of Biomedical Engineering, Johns Hopkins University, Baltimore, Maryland, United States of America, 4 Department of Pathology, Johns Hopkins School of Medicine, Baltimore, Maryland, United States of America, 5 Departments of Computer Science and Biostatistics, Johns Hopkins University, Baltimore, Maryland, United States of America, 6 Division of Emergency Medicine, Johns Hopkins School of Medicine, Baltimore, Maryland, United States of America, 7 Department of Molecular Microbiology and Immunology, Johns Hopkins University Bloomberg School of Public Health, Baltimore, Maryland, United States of America, 8 Infectious Disease Institute, Makerere University College of Health Sciences, Kampala, Uganda

☺ These authors contributed equally to this work.
‡ These authors are joint senior authors on this work. YCM and AJK also contributed equally to this work.
* abrahamjk@jhmi.edu

## Abstract

### Objectives

Accurate diagnosis of existing and emerging respiratory pathogens is important. We evaluated the capability of unbiased metagenomic next generation sequencing (mNGS) to identify pathogenic RNA viruses from two cohorts of nasopharyngeal (NP) swabs previously tested by commercial multiplex respiratory diagnostics.

### Methods

NP swabs (N = 100) in viral transport media (VTM) were assessed using mNGS for this study. Cohort 1 (N = 52) consisted of symptomatic individuals who tested negative for SARS-CoV-2, influenza A/B, and RSV by the Xpert Xpress CoV-2/Flu/RSV Plus multiplex respiratory virus panel and were tested by mNGS for undetected pathogens. Cohort 2 (N = 48) included symptomatic individuals who were positive (N = 26) or negative (N = 22) by the ePlex RP2 multiplex respiratory pathogen panel. Samples were positive for influenza A (N = 8), rhinovirus/enterovirus (N = 5), RSV (N = 4), adenovirus (N = 3), parainfluenza (N = 2), seasonal coronaviruses (N = 2), and human metapneumovirus (N = 1), as well as a rhinovirus/enterovirus/human metapneumovirus co-infected sample (N = 1). mNGS results were compared with ePlex RP2

**Data availability statement:** The metagenomic data set generated and analyzed during this study has been deposited in the NCBI Sequence Read Archive with accession code PRJNA1453162 (https://www.ncbi.nlm.nih.gov/bioproject/PRJNA1453162).

**Funding:** 1) SS:National Institutes of Health (NIH R35-GM130151) https://grants.nih.gov 2) YCM:Johns Hopkins Center of Excellence for Influenza Research and Response (HHS 75N93021C00045, HHS N272201400007C) https://www.ceirr-network.org/centers/jh-ceirr The funders played no role in the study design, data collection and analysis, decision to publish, or preparation of the manuscript?

**Competing interests:** The authors have declared that no competing interests exist.

findings, and symptomatic negative samples were evaluated for additional pathogen detection.

## Results

Cohort 1 contained 8% (4/52) viral and 19% (10/52) bacterial reads. In cohort 2, positive concordance between ePlex RP2 and mNGS was 31% (8/26). mNGS did not identify any viral reads in ePlex RP2-negative samples. However, it detected other microbial reads, such as *Acanthamoeba castellanii,* in 21% (10/48) of samples.

## Conclusion

In this study, targeted multiplex amplification methods demonstrated better overall sensitivity in NPs of symptomatic respiratory individuals than mNGS. Other mNGS approaches may produce different results. This study suggests that mNGS may offer adjunctive information, including the detection of rare pathogens, which may be helpful in some clinical contexts.

## Introduction

Respiratory viruses are common causes of upper and lower respiratory tract illness with resulting significant morbidity and mortality [1–3]. A recent report from the US Centers for Disease Control and Prevention for 2024−2025 showed increasing hospitalization rates largely driven by influenza and SARS-CoV-2 among older adults and RSV in children [4]. Furthermore, respiratory infections are increasingly associated with microbial co-infections, which have been associated with increased disease severity [1,2]. In such instances, it is important that the diagnostic pipeline can detect multiple pathogens rapidly to allow for timely treatment and the introduction of appropriate infection control interventions to reduce the spread of infections.

Syndromic respiratory viral and bacterial multiplex panels have been developed to enable the detection of more than 20 predefined respiratory pathogens with high sensitivity and specificity [3,5]. Despite these large panel sizes, only 30–35% of patients with influenza-like illness have an identifiable pathogen [6]. These assays represent an improvement over traditional methods such as viral culture and direct fluorescent antibody testing and have received U.S. Food and Drug Administration (FDA) clearance based on their performance. Other approaches include sequencing-based probe panels and targeted next-generation sequencing. These approaches have improved the ability to detect a broader range of microbial pathogens than multiplex PCR assays, however, they are limited to predefined genomes, can miss detection of novel, reemerging, or divergent pathogens as highlighted during the COVID-19 pandemic [7], and are largely not FDA approved. In such instances, unbiased metagenomic next generation sequencing (mNGS) can be advantageous by allowing an agnostic approach [8–10]. mNGS has become popular with clinicians as an unbiased, hypothesis-free genomic interrogation of clinical specimens, and as the costs associated with sequencing decrease, mNGS is an increasingly attractive

way to diagnose the etiologic cause of infection. However, many of these methods do not have established performance characteristics, such as sensitivity and specificity. Previous work from our group using plasma samples has shown that mNGS can consistently detect viral nucleic acids, albeit at concentrations ≥ $10^4$/mL compared with clinical assays [8,11]. Metagenomic approaches using nasopharyngeal (NP) swabs have not been evaluated relative to current multiplex PCR approaches that are the mainstay of most clinical laboratories for respiratory virus detection.

We sought to apply mNGS to the evaluation of NP swabs in viral transport media (VTM) from two symptomatic cohorts, both initially tested with commercial multiplex molecular diagnostics, to test the concordance of mNGS detections against these commercial diagnostics, and to assess whether mNGS would give additional detection 'calls' for symptomatic multiplex PCR-negative participants given the high levels of sequence variations among respiratory viruses and continued emergence of novel viruses.

## Methods

### Study design

This cross-sectional cohort study evaluated symptomatic study participants, using a convenience sample of NP swabs from two distinct cohorts (n = 100). The median age of the participants across both cohorts was 40 years (IQR 31−51) with 68% females. Cohort 1 (N = 52) was collected during the 2020−21 SARS-CoV-2 pandemic from symptomatic individuals who tested negative for SARS-CoV-2, Influenza A/B, and RSV using the Xpert Xpress CoV-2/Flu/RSV Plus (Cepheid, Sunnyvale, CA). Cohort 2 (N = 48) included NP swabs collected during the 2017−18 and 2018−2019 respiratory virus seasons from symptomatic individuals with influenza-like illness and were evaluated by ePlex RP2. These samples were a mix of positives (N = 26) and negatives (N = 22), with positive samples consisting of influenza A (N = 8), rhinovirus/enterovirus (N = 5), RSV (N = 4), adenovirus (N = 3), parainfluenza (N = 2), seasonal coronaviruses (N = 2), human metapneumovirus (N = 1) and lastly a rhinovirus/enterovirus/human metapneumovirus co-infected sample (N = 1). Participant characteristics of the two cohorts along with ePlex RP2 results are reported in Table 1. Residual VTM was aliquoted and stored at −80⁰c until testing with mNGS. All samples underwent identical processing for this study. Informed consent was obtained from all study participants, and the study was approved by the Johns Hopkins Institutional Review Board (IRB00091667, IRB00306448). Study samples and results were accessed on March 2,2023, and de-identified demographic data were accessed on Aug 15,2023.

### Xpert Xpress CoV-2/Flu/RSV Plus

*The* Xpert Xpress CoV-2/Flu/RSV Plus is a cartridge-based, assay used for the detection of a panel of respiratory viruses [12]. Testing was performed per manufacturer instructions and included the following organisms: influenza A H1, influenza A H3, influenza B, SARS-CoV-2, and RSV A/B. Briefly, 300 μl of sample was added to an Xpert Xpress CoV-2/Flu/RSV Plus cartridge, the cartridge was closed and analyzed on the Xpert instrument. Sample results were recorded directly from the instrument.

### ePlex RP2

The ePlex RP2 is a cartridge-based multiplex nucleic acid amplification assay designed for syndromic detection of respiratory viral and bacterial pathogens. It can detect more microbial pathogens than Xpert Xpress CoV-2/Flu/RSV Plus**.** Analysis was performed per manufacturer instructions utilizing ePlex RP2 panel cartridges, which include the following organisms: adenovirus, seasonal coronaviruses (229E, HKU1, NL63, and OC43), SARS-CoV-2, humanmetapneumovirus, rhinovirus/enterovirus, influenza A H1, influenza A H3, influenza B, parainfluenza viruses 1–4, RSV A/B *Chlamydia pneumoniae*; and *Mycoplasma pneumoniae* [13]. Briefly, 200 μl of sample was added to an RP2 cartridge, the cartridge was closed and analyzed on the ePlex instrument. Sample results were recorded from the instrument.

**Table 1. Participant characteristics and ePlex RP2 results.**

| Category | Total Study (N=100) Percentage (N/N) | Cohort 1 (N=52) Percentage (N/N) | Cohort 2 Positives (N=26) Percentage (N/N) | Cohort 2 Negatives (N=22) Percentage (N/N) |
|---|---|---|---|---|
| **Age Range (Years)** | | | | |
| 18-24 | 15% (15/100) | 15% (8/52) | 15% (4/26) | 14% (3/22) |
| 25-34 | 20% (20/100) | 25% (13/52) | 12% (3/26) | 18% (4/22) |
| 35-44 | 27% (27/100) | 27% (14/52) | 23% (6/26) | 32% (7/22) |
| 45-54 | 22% (22/100) | 21% (11/52) | 23% (6/26) | 23% (5/22) |
| >55 | 16% (16/100) | 12% (6/52) | 27% (7/26) | 14% (3/22) |
| **Gender** | | | | |
| Male | 32% (32/100) | 35% (18/52) | 27% (7/26) | 32% (7/22) |
| Female | 67% (67/100) | 63% (33/52) | 73% (19/26) | 68% (15/22) |
| No Answer | 1% (1/100) | 2% (1/52) | NA[a] | NA |
| **BMI** | | | | |
| 18-24 | 12% (12/100) | NA | 15% (4/26) | 36% (8/22) |
| 25-34 | 23% (23/100) | NA | 50% (13/26) | 45% (10/22) |
| 35-44 | 5% (5/100) | NA | 12% (3/26) | 9% (2/22) |
| 45-54 | 5% (5/100) | NA | 19% (5/26) | NA |
| Not reported | 55% (55/100) | 100% (52/52) | 4% (1/26) | 9% (2/22) |
| **Hospitalization** | | | | |
| Yes | 1% (1/100) | NA | 4% (1/26) | NA |
| No | 47% (47/100) | NA | 96% (25/26) | 100% (22/22) |
| Not Reported | 52% (52/100) | 100% (52/52) | NA | NA |
| **Antibiotics[b]** | | | | |
| Yes | 8% (8/100) | NA | 8% (2/26) | 27% (6/22) |
| No | 40% (40/100) | NA | 92% (24/26) | 73% (16/22) |
| Not Reported | 52% (52/100) | 100% (52/52) | NA | NA |
| **Antivirals[b]** | | | | |
| Yes | 3% (3/100) | NA | 4% (1/26) | 9% (2/22) |
| No | 45% (45/100) | NA | 96% (25/26) | 91% (20/42) |
| Not Reported | 52% (52/100) | 100% (52/52) | NA | NA |

a–Not applicable; b-Used within past 30 days; cohort 1 tested with Xpert Xpress Xpress CoV-2/Flu/RSV; Cohort 2 ePlex RP2.

## Sample preparation and nucleic acid extraction

The extraction of DNA and RNA was performed as previously described [8]. The ZR-Duet DNA/RNA miniprep (Cat#D7003, Zymo Research) enables the extraction of purified RNA and DNA extraction as separate fractions, thereby reducing DNA contamination in the RNA extract while also allowing for analysis of purified DNA. However, for this study, only the RNA fraction was utilized for library preparation and sequencing, as the metagenomic workflow was optimized for microbial RNA detection.

Briefly, the samples from each study participant were initially centrifuged at 1,600g for 15 minutes at 4°C and then syringe filtered using 0.2µm (Cat #WHA67781302, Whatman®, Millipore Sigma). Following filtration, 200µl was used for extraction of DNA and RNA following the kit protocol and eluted in 60µl final volume.

Xpert Xpress CoV-2/Flu/RSV Plus and ePlex RP2 are FDA-authorized assays that include internal cartridge controls to ensure proper assay performance. In contrast, mNGS lacks standardized controls; therefore, a commercial positive control (Acrometrix™ Multi-Analyte SARS-CoV-2, influenza A/B, and RSV A/B), containing approximately $10^4$ copies/

mL of each viral target was included to assess the mNGS workflow. These were assigned as controls 1–4. In parallel, an in-house SARS-CoV-2–positive de-identified NP sample, quantified by droplet digital PCR at $10^8$ copies/mL, was used to generate a dilution series ranging from $10^6$ to $10^1$ copies/mL to assess the limit of detection. In the absence of identical controls across platforms, findings were interpreted based on qualitative detection.

### Next generation sequencing

mNGS testing was performed using the Revelo RNA-Seq kit (Cat#30201358, Tecan Genomics). Briefly, 10 µl of extracted RNA underwent strand-specific RNA-seq library preparation using the Revelo RNA-Seq (#30201358, Tecan Genomics). The libraries were pooled based on the NuQuant library quantification provided. Pooled libraries were sequenced with an Illumina Novaseq 6000 (Illumina, Inc., CA, USA) to obtain 100 basepair (bp) paired end reads (~800 million single reads).

### Taxonomic classification of reads

Metagenomics read outputs were analyzed using KrakenUniq and the Kraken protocol as previously described [8]. For this study, human reads were filtered utilizing Bowtie2 and T2T CHM13 human genome as a reference. The remaining host-filtered reads were then classified by KrakenUniq against a customized database built from all RefSeq complete bacterial, viral, archaeal genomes, a curated collection of eukaryotic pathogen genomes from EuPathDB, the UniVec set of standard laboratory vectors from NCBI, and the GRCh38 human genome. For a microbial species to be present in the specimen a cutoff of ≥50 classified reads was used. For subsequent analysis, KrakenUniq reports were loaded into Pavian [14] which provides sample visualizations, read count comparisons, and z-score calculations. A z-score statistical cutoff was used to establish a true species call with the smallest number of reads satisfying this cutoff in any sample was 5. All software tools used in this study are linked and available for download at http://ccb.jhu.edu/data/kraken2_protocol/.

## Results

### mNGS of Xpert Xpress CoV-2/Flu/RSV Plus negative samples

Except for influenza B, all viruses present in the Acrometrix™ Multi-Analyte SARS-CoV-2, influenza A/B, and RSV A/B control, which were included as controls 1 and 2 with cohort 1 samples, were successfully detected. SARS-CoV-2 reads were also detected in all dilutions ($10^6$ to $10^1$ copies/mL) (S1 Fig). Library preparation of samples from cohort 1 generated a median of 15,503,069 reads (IQR 14,392,221), of which a median of 13,766,821 reads (IQR 10,362,267) were classified as non-human reads (S1 Table). Among samples testing negative by the Xpert Xpress CoV-2/Flu/RSV Plus assay, mNGS identified microbial sequences (defined here as non-human sequences originating from viruses, bacteria, fungi, or parasites) in 27% (14/52), including RSV (n = 1) and rhinovirus (n = 3). Bacterial species were detected in 19% (10/52) of samples.

### Comparison of samples tested with ePlex RP2 and mNGS

Not all viral targets included in the Acrometrix™ Multi-Analyte SARS-CoV-2, influenza A/B, and RSV A/B control, included as controls 3 and 4 with cohort 2 samples, were detected. Specifically, influenza A and B were not detected, whereas SARS-CoV-2 and RSV were detected (Fig 1b). A median of 23,937,404 reads (IQR 10,713,220) were generated per sample in cohort 2. Of these, a median of 22,919,075 reads (IQR 10,999,984) were classified as non-human (S1 Table) and analyzed for microbial reads. Results for cohort 2 comparing mNGS and ePlex RP2 (n = 48) are summarized in S2 Table. Positive concordance between ePlex RP2 and mNGS was 31% (8/26) overall, with 16 samples testing negative by both methodologies. Additionally, mNGS identified pathogenic or potentially pathogenic bacterial species, in 21% (10/48) of samples (S2 Table). Notably, mNGS detected *Acanthamoeba castellanii* sequences in two samples. The taxonomical classification with read counts is shown in Fig 1b. In this report, the term "potentially pathogenic" is used to

a)

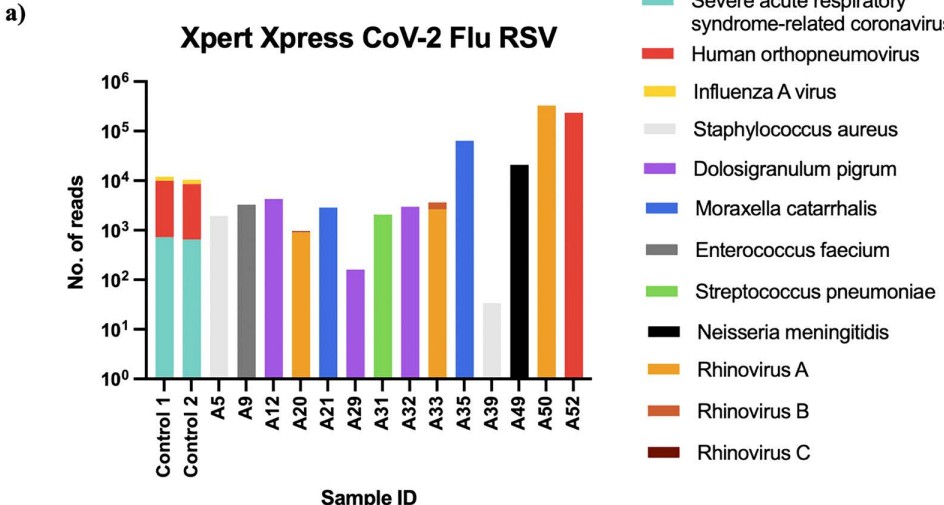

b)

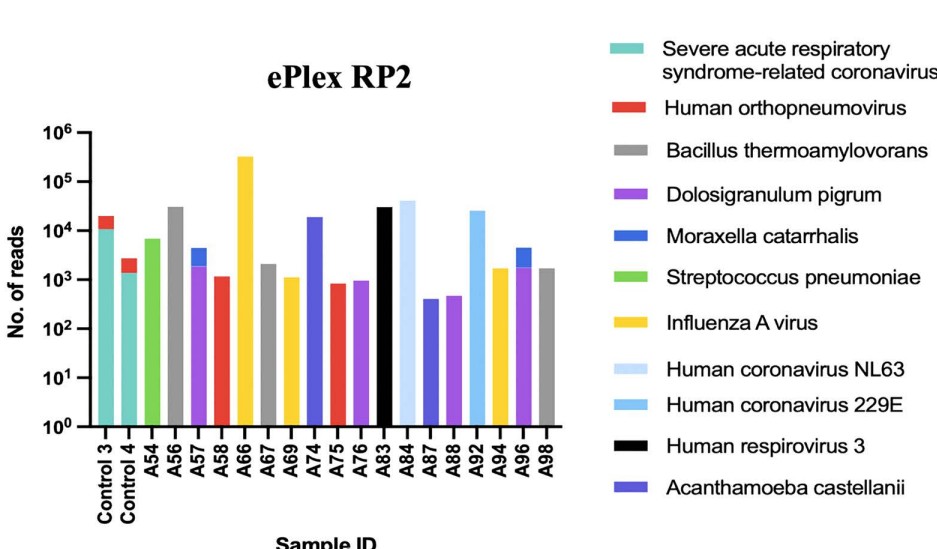

**Fig 1. Microbial reads detected by mNGS in nasopharyngeal (NP) swabs from two cohorts.** Number of microbial reads detected by mNGS in NP swabs previously tested with **(a)** Xpert Xpress CoV-2/Flu/RSV and **(b)** ePlex RP2 panel. Controls 1–4: Acrometrix™ Multi-Analyte containing $10^4$ copies/mL of SARS-CoV-2, Flu A/B, and RSV A/B. The x-axis denotes individual samples with microbial reads classified taxonomically, and the y-axis denotes the total number of microbial reads per sample. Human orthopneumovirus refers to respiratory syncytial virus (RSV) subtypes A and B.

describe organisms that may cause disease only under specific circumstances, such as host conditions. In the absence of supporting confirmatory or clinical evidence, it is difficult to draw a definitive conclusion about their role in disease etiology. Genome-level read counts and coverage for selected microbial sequences using the Pavian alignment viewer are shown in Fig 2. The Pavian alignment viewer allows visualization of genome-level read coverage across reference genomes, allowing assessment of read distribution and localized coverage patterns that may indicate contamination or misclassification.

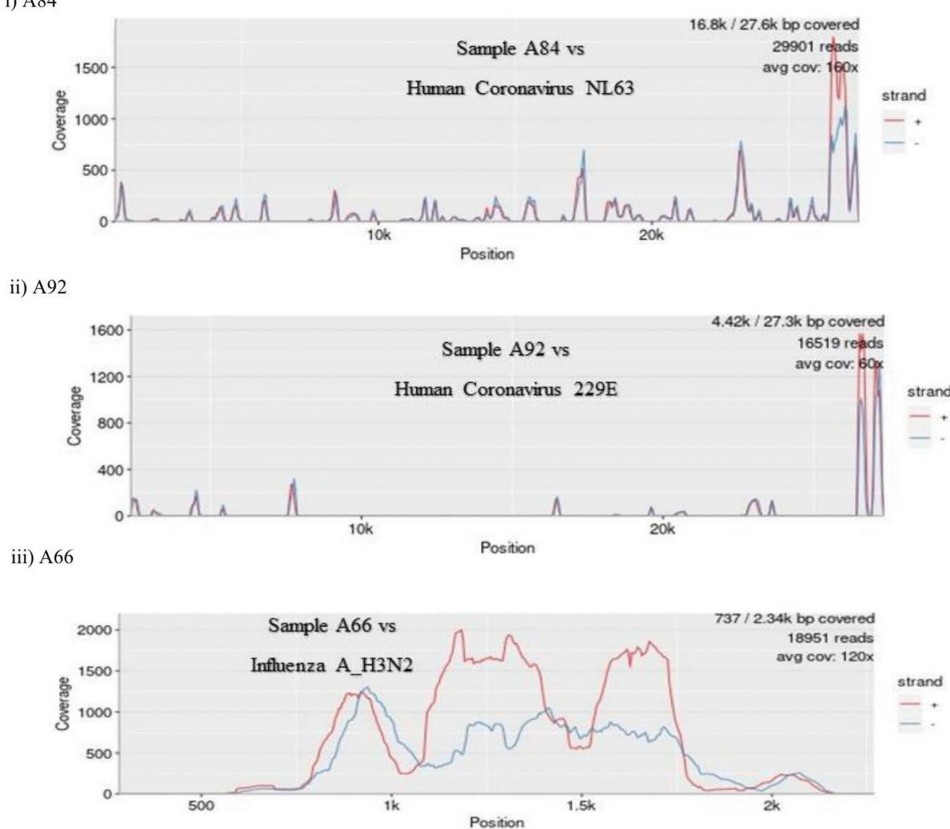

i) A84

ii) A92

iii) A66

**Fig 2. Pavian visualization showing genome-level coverage plots for selected pathogens identified by mNGS.** The alignment plot illustrates the genome coverage of three detected pathogens mapped to the respective reference sequences in three study samples. **i)** Human coronavirus NL63 (Ref-Seq assembly: GCF_000853865.1), **ii)** Human Coronavirus 229E (RefSeq assembly: GCF_000853505.1), and **iii)** Influenza A H3N2 (RefSeq assembly: GCF_000865085.1). The x-axis denotes genomic position, and y-axis denotes the read coverage.

## Discussion

mNGS was employed to detect respiratory viruses in NP swab samples from symptomatic participants that had previously undergone testing using either the Xpert Xpress CoV-2/Flu/RSV Plus (n = 52) or the ePlex RP2 panel (n = 48). mNGS has been previously employed for diagnosis of infectious pathogens, with a recent meta-analysis reporting a pooled sensitivity of 75% and specificity of 68% [9]. The diagnostic performance, as assessed by the area under the receiver operating characteristic curve (AUC), was estimated at 85% [9]. Other reports have indicated suitability for respiratory virus surveillance, using different sequencing methodologies and sample types [10,15], including one methodology receiving breakthrough designation by the Food and Drug Administration (FDA) for NP swabs and bronchoalveolar lavage (BAL) [15]. Targeted sequencing panels have produced similar results, indicating their suitability for analyzing clinical samples [16,17]. However, our results indicated an overall positive concordance with ePlex RP2 testing of 31% (8/26) overall, which is markedly lower than previous reports [9,10,15]. mNGS did detect pathogenic microbial species not included in the ePlex RP2 test menu (e.g., *Acanthamoeba castellanii*). Among samples that tested negative by the more limited Xpert Xpress CoV-2/Flu/RSV Plus panel, 8% (4/52) yielded identifiable viral reads that were either missed or could not be detected by the panel, alongside the detection of bacterial species which are not included in the Xpert Xpress CoV-2/Flu/RSV Plus panel. For the bacterial taxa identified, it remains unclear whether these represent true pathogenic infections or reflect high-level colonization.

While the applicability of mNGS for infectious diseases has shown promise based on previous studies [8,9,15,18]. Our findings might also reflect some of the limitations of mNGS for detection of RNA viruses, which require extra steps in sample preparation prior to sequencing, has a high host nucleic acid background, and requires complex analysis to validate detections. We have previously shown in our study among individuals with acute febrile illness that mNGS protocol in plasma failed to identify HIV-1 present at a concentration <4 $\log_{10}$ copies/ml [8]. In another study viremia levels detected by qPCR were <4 $\log_{10}$ copies or IU/mL, there was low correlation (r = 0.28) between mNGS and qPCR [11]. This may partially explain the discordance between ePlex RP2 and mNGS. ePlex RP2 does not generate quantitative data which also complicates direct comparisons between methods. Additionally, while ePlex RP2 testing was performed on freshly collected specimens (2017–2019), mNGS analysis was conducted retrospectively on archived frozen samples. The extent to which RNA degradation in VTM over time may impact detection sensitivity remains unclear [19,20]. Other limitations include the sample volume processed for mNGS (200 μl), as processing larger sample volumes could increase the likelihood of sequencing low copy viral transcripts in the samples tested. Lastly, we only performed unbiased mNGS on these samples; utilization of one of the targeted sequencing panels [16,17] that are commercially available could have produced different results.

Overall, mNGS did not identify any viral reads in ePlex RP2-negative samples. It should be noted that we employed unbiased mNGS as a research method and did not apply a sequencing methodology previously validated for detection of respiratory pathogens. The methodology we used, however, demonstrated added value in detecting viral pathogens missed by the more limited Xpress CoV-2/Flu/RSV Plus assay in symptomatic respiratory participants. Specific PCR-based detection should be employed before resorting to mNGS which is time and labor intensive and lacks sensitivity. Due to its pathogen agnostic detection approach, mNGS retains a critical role in scenarios where acute infection with a novel, emerging, or rare pathogen is suspected, underscoring its importance in specialized diagnostic investigations.

## Supporting information

**S1 Fig. Detection of SARS-CoV-2 reads across the dilution series.**
(DOCX)

**S1 Table. Total and non-human read counts for controls and samples from cohort 1 and cohort 2 sequenced on the NovaSeq 6000.**
(DOCX)

**S2 Table. mNGS results for samples evaluated by ePlex RP2.**
(DOCX)

## Author contributions

**Conceptualization:** Yukari C Manabe, Abraham J Kandathil.

**Data curation:** Justin Hardick, Raghavendran Anantharam, Jennifer Lu, Richard E. Rothman, Katherine ZJ Fenstermacher, Andrew Pekosz, Annet Onzia, Lydia Nakiyingi.

**Funding acquisition:** Steven L Salzberg, Richard E. Rothman, Andrew Pekosz, Yukari C Manabe.

**Investigation:** Justin Hardick, Raghavendran Anantharam, Jennifer Lu, Richard E. Rothman, Katherine ZJ Fenstermacher, Andrew Pekosz, Annet Onzia, Lydia Nakiyingi.

**Methodology:** Justin Hardick, Raghavendran Anantharam, Jennifer Lu, Steven L Salzberg, Andrew Pekosz, Yukari C Manabe, Abraham J Kandathil.

**Software:** Jennifer Lu, Steven L Salzberg.

**Supervision:** Steven L Salzberg, Yukari C Manabe, Abraham J Kandathil.

**Writing – original draft:** Justin Hardick, Raghavendran Anantharam, Jennifer Lu, Steven L Salzberg, Andrew Pekosz, Yukari C Manabe, Abraham J Kandathil.

**Writing – review & editing:** Justin Hardick, Raghavendran Anantharam, Steven L Salzberg, Andrew Pekosz, Yukari C Manabe, Abraham J Kandathil.

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
