## [Decision Letter · Decision Letter 0]

18 Jan 2026

PONE-D-25-54639Title: Comparison of unbiased Metagenomic Next Generation Sequencing to Targeted Multiplex Diagnostic Assays for the Detection of Respiratory VirusesPLOS One

Dear Dr. Kandathil,

Thank you for submitting your manuscript to PLOS ONE. After careful consideration, we feel that it has merit but does not fully meet PLOS ONE’s publication criteria as it currently stands. Therefore, we invite you to submit a revised version of the manuscript that addresses the points raised during the review process.

We look forward to receiving your revised manuscript.

Kind regards,

Guocan Yu

Academic Editor

PLOS One

Journal Requirements:

3. In the online submission form, you indicated that your data will be submitted to a repository upon acceptance.  We strongly recommend all authors deposit their data before acceptance, as the process can be lengthy and hold up publication timelines. Please note that, though access restrictions are acceptable now, your entire minimal  dataset will need to be made freely accessible if your manuscript is accepted for publication. This policy applies to all data except where public deposition would breach compliance with the protocol approved by your research ethics board. If you are unable to adhere to our open data policy, please kindly revise your statement to explain your reasoning and we will seek the editor's input on an exemption.

4. Please be informed that funding information should not appear in the Acknowledgments section or other areas of your manuscript. We will only publish funding information present in the Funding Statement section of the online submission form. Please remove any funding-related text from the manuscript.

Reviewers' comments:

Reviewer's Responses to Questions

**Comments to the Author**

1. Is the manuscript technically sound, and do the data support the conclusions?

Reviewer #1: Partly

Reviewer #2: Partly

2. Has the statistical analysis been performed appropriately and rigorously? 

Reviewer #1: N/A

Reviewer #2: No

3. Have the authors made all data underlying the findings in their manuscript fully available?

Reviewer #1: No

Reviewer #2: No

4. Is the manuscript presented in an intelligible fashion and written in standard English?

Reviewer #1: No

Reviewer #2: Yes

5. Review Comments to the Author

Reviewer #1: Comments:

The manuscript reports a comparison between two diagnostic approaches: multiplex PCR-based viral panels and metagenomic NGS, applied to two relatively small cohorts of approximately 50 samples each. It’s not clear why the authors chose specifically to compare between these two approaches, but completely neglect to mention sequencing-based probe panel approaches like TWIST, Illumina Viral Surveillance Panel v2, or the earlier ViroCap, which target a broader range of viruses than multiplex PCR panels, while being more focused and specific than untargeted metagenomic sequencing. This important category which is gaining popularity as a diagnostic assay is entirely overlooked, including in the introduction and discussion. Overall the manuscript is not written well or very clear. The text is brief and lacks sufficient detail, making it difficult to follow. The current format resembles a short communication or brief report, and many parts are not clearly articulated. In several sections, explanations are omitted and substantial prior knowledge is assumed, which considerably impairs readability and interpretation.

The manuscript would benefit from a thorough rewrite to improve clarity, expand on methodological details, and provide more explicit explanations of the results and their implications.

Other comments:

• Abstract:

- I would mention what Xpert Xpress / ePlex RP2 are, it’s not very clear.

- this part “Participant median age was 40 years (IQR 31-51) with 68% (68/100) females” is more methods than results, and maybe shouldn’t be included in the abstract.

- Its not clear from the abstract that the Xpert Xpress (cohort 1) are the “negatives” that were evaluated with mNGS.

• Line 47: fix the wording of the sentence

• Line 57: sentence not so clear - what other samples? Is this concentration good? Compared to what?

• Line 70: missing details about the ePlex RP2 system.

• Line 70: “clinical characteristics..” add “and ePlex RP2 test results” (also to the table header)

• Line 86: define which of the controls you refer to later on as “control 1” and “control 2” (e.g. in Figure 1a)

• Line 95: how many reads were allocated per sample?

• Line 107: why was the >50 reads threshold chosen? (out of how many reads?) is this a Kraken threshold, or a custom one?

• Line 118: “some of the detected species are potentially pathogenic” – not clear what this means? By what measure?

• Line 121: I would phrase a different, more informative header, maybe mentioning the type of samples shown in the figure and the methods used rather than the tool used to identify them.

• Line 122: use mNGS acronym

• Figure 1a: control 1/2 finding should be mentioned in the text (and defined better, also in methodology). E.g., it’s not clear to all readers that RSV, that’s included in the controls is a member of the Orthopneumovirus family. Also, it says in the figure legend “control 1-4” – that’s not clear. In addition, it says that just the 14 samples that were positive for bacteria are shown, but the figure legend says also “Rhinovirus” which is a virus. So, need to be more specific and clarify this in the text.

• Figure 2 is not explained very well in the results section. Need to clarify what it shows and why it is important to show it as a main figure.

• Supp figure 1: not clear what e1-e6 is.

• Line 144: the “controls” sub section can be integrated into the main text, each part where it fits, its missing from there (see my previous comments).

Reviewer #2: Table 01 – data not presented in a clear, legible manner. Why was data on race, ethnicity collected? Was there any analysis done relating to these factors? If not, does it add anything of value to the analysis?

Methods – the nucleic acid extraction mentions both RNA and DNA, but the library preparation only deals with RNA seq. was only RNA analyzed? In that case why was DNA also extracted?

Controls – there seems to be a limitation on detection of Influenza A/B from Acrometrix positive control. Was this also tested on the Xpert Xpress/ePlex RP2 to check whether the targetted panels could detect as expected?

Sequencing - can benefit from more detail on how much data per sample - this is a parameter that directly impacts limit of detection.

The details on how sensitivity/specificity was calculated is unclear. Would benefit from a specificity/sensitivity table.

Overall conclusion needs more justification – you are saying NGS has high limit of detection but there is no data showing the ePlex and Xpert were run on the same controls. Using the clinical data for justification is difficult as there are variables such as storage time that may have impacted viability of RNA.

There should be more detail on the limitation of the study methodology, the NGS process etc. There is no analysis in to the possible factors that may be affecting the detection or output on the NGS process.

Figure 01 – It is not clear from the figure title that it is showing NGS data from a specific cohort.

Supplemental file – only ePlex detailed data is shared. Ideally the Xpert breakdown should also be included.

6. PLOS authors have the option to publish the peer review history of their article (what does this mean?). If published, this will include your full peer review and any attached files.

Reviewer #1: No

Reviewer #2: No

---

## [Author Response · Author response to Decision Letter 1]

11 Mar 2026

We thank the reviewers for their comments and have included our responses in a separate attachment with the submission.

---

## [Decision Letter · Decision Letter 1]

7 Apr 2026

Title: Comparison of unbiased Metagenomic Next Generation Sequencing to Targeted Multiplex Diagnostic Assays for the Detection of Respiratory Viruses

PONE-D-25-54639R1

Dear Dr. Abraham J Kandathil,

We’re pleased to inform you that your manuscript has been judged scientifically suitable for publication and will be formally accepted for publication once it meets all outstanding technical requirements.

Kind regards,

Guocan Yu

Academic Editor

PLOS One

Additional Editor Comments (optional):

Reviewers' comments:

Reviewer's Responses to Questions

**Comments to the Author**

1. If the authors have adequately addressed your comments raised in a previous round of review and you feel that this manuscript is now acceptable for publication, you may indicate that here to bypass the “Comments to the Author” section, enter your conflict of interest statement in the “Confidential to Editor” section, and submit your "Accept" recommendation.

Reviewer #1: All comments have been addressed

2. Is the manuscript technically sound, and do the data support the conclusions?

Reviewer #1: Yes

3. Has the statistical analysis been performed appropriately and rigorously? 

Reviewer #1: N/A

4. Have the authors made all data underlying the findings in their manuscript fully available?

Reviewer #1: Yes

5. Is the manuscript presented in an intelligible fashion and written in standard English?

Reviewer #1: Yes

6. Review Comments to the Author

Reviewer #1: the authors have addressed all the comments listed in the review and as a result the manuscript is now improved.

7. PLOS authors have the option to publish the peer review history of their article (what does this mean?). If published, this will include your full peer review and any attached files.

Reviewer #1: No

---

## [Editor Report · Acceptance letter]

PONE-D-25-54639R1

PLOS One

Dear Dr. Kandathil,

I'm pleased to inform you that your manuscript has been deemed suitable for publication in PLOS One. Congratulations! Your manuscript is now being handed over to our production team.

Kind regards,

on behalf of

Dr. Guocan Yu

Academic Editor

PLOS One